# Estimation of country-level basic reproductive ratios for novel Coronavirus (SARS-CoV-2/ COVID-19) using synthetic contact matrices

**Joe Hilton** [1,2,3]*, **Matt J. Keeling** [1,2,3]

**1** School of Life Sciences, University of Warwick, Coventry, United Kingdom, **2** Zeeman Institue (SBIDER), University of Warwick, Coventry, United Kingdom, **3** Mathematics Institute, University of Warwick, Coventry, United Kingdom

* j.hilton@warwick.ac.uk

**Data Availability Statement:** All of the data and code used, as well as the outputs generated, are

## Abstract

The 2019-2020 pandemic of atypical pneumonia (COVID-19) caused by the virus SARS-CoV-2 has spread globally and has the potential to infect large numbers of people in every country. Estimating the country-specific basic reproductive ratio is a vital first step in public-health planning. The basic reproductive ratio ($R_0$) is determined by both the nature of pathogen and the network of human contacts through which the disease can spread, which is itself dependent on population age structure and household composition. Here we introduce a transmission model combining age-stratified contact frequencies with age-dependent susceptibility, probability of clinical symptoms, and transmission from asymptomatic (or mild) cases, which we use to estimate the country-specific basic reproductive ratio of COVID-19 for 152 countries. Using early outbreak data from China and a synthetic contact matrix, we estimate an age-stratified transmission structure which can then be extrapolated to 151 other countries for which synthetic contact matrices also exist. This defines a set of country-specific transmission structures from which we can calculate the basic reproductive ratio for each country. Our predicted $R_0$ is critically sensitive to the intensity of transmission from asymptomatic cases; with low asymptomatic transmission the highest values are predicted across Eastern Europe and Japan and the lowest across Africa, Central America and South-Western Asia. This pattern is largely driven by the ratio of children to older adults in each country and the observed propensity of clinical cases in the elderly. If asymptomatic cases have comparable transmission to detected cases, the pattern is reversed. Our results demonstrate the importance of age-specific heterogeneities going beyond contact structure to the spread of COVID-19. These heterogeneities give COVID-19 the capacity to spread particularly quickly in countries with older populations, and that intensive control measures are likely to be necessary to impede its progress in these countries.

available at https://github.com/JBHilton/hilton-keeling-estimating-R0.

**Funding:** This research was funded by: the National Institute for Health Research (NIHR) (project reference 17/63/82) using UK aid from the UK Government to support global health research JH & MJK; the Engineering and Physical Sciences Research Council (EPSRC) (grant reference EP/S022244/1) MJK; Health Data Research UK, which receives its funding from HDR UK Ltd (NIWA1) MJK. The views expressed in this publication are those of the authors and not necessarily those of the any of the funders. The funders had no role in study design, data collection and analysis, decision to publish, or preparation of the manuscript.

**Competing interests:** The authors have declared that no competing interests exist.

## Author summary

Over 100 countries have reported laboratory-confirmed cases of atypical pneumonia caused by 2019 novel coronavirus (COVID-19). Cases are largely reported in older age groups, suggesting a strong age-dependent component to either transmission or the probability of developing symptoms and thus being detected. We introduce a mathematical model for COVID-19 transmission in which contact behaviour, susceptibility, detection probability, and transmission from undetected cases all vary with age. We fit our model to epidemiological data from the outbreak in China for the special case where asymptomatic transmission is negligible, and compare it to a null model where only contact behaviour varies with age. Our fitted model suggests that contacts involving older individuals are particularly likely to generate new detected cases, intensifying the spread of infection in countries with older populations. We estimate the basic reproductive ratio (a measure of a pathogen's capacity for spread) of COVID-19 in 152 countries under both models, and find that estimates of the basic reproductive ratio are highly dependent on the assumed underlying transmission structure; our more complex model predicts higher values in Japan and much of Europe and lower values in much of Africa, in comparison to the contact frequency-based model where this pattern is reversed.

## Introduction

The ongoing epidemic of atpyical pneumonia (COVID-19) emerged in Wuhan, China in late 2019, with the novel coronavirus SARS-CoV-2 identified as its causative agent [1]. It exhibited an early capacity for global spread [2], and, as of May 29th 2020, it is present on all inhabited continents and has spread to 216 countries and territories [3]. A surprising feature of the outbreak in all countries is that the distribution of detected cases is characterised by large numbers of cases in older individuals and fewer in younger individuals, with particularly low numbers in under-15s [1, 4]. Unfortunately, it remains unclear what role these younger age-groups play in onward transmission.

The basic reproductive ratio, commonly denoted $R_0$, is a key epidemiological variable in any outbreak. It is defined as the expected number of infectious cases generated by a single average case in an entirely susceptible population [5, 6]. It is therefore an important measure of an infection's capacity to spread, with sustained transmission possible only when $R_0 > 1$, and can be used to calculate both the expected size of an outbreak and the threshold level of vaccination necessary for eradication [5]. Early estimates of $R_0$ for COVID-19 in China have mostly clustered between 2 and 4 [1, 2, 7–11]. However, these estimated values of $R_0$ may not necessarily reflect the intensity of spread in countries other than China; this is because the basic reproductive ratio is dependent on both pathogen and host population characteristics [12].

Age-structured infectious disease models have been hugely successful in describing the dynamics of a number of infectious diseases, including endemic infections and novel outbreaks [13–17]. These models typically use a so-called "who acquires infection from whom" paradigm, with transmission rates between individuals in different age classes encoded either in a single matrix or in a set of matrices specific to transmission events in different locations [5, 13]. It is often assumed that age-structured transmission rates are directly proportional to the frequency and duration of age-structured contacts, measured from social contact studies [15, 16, 18]. Such studies typically find that contact intensities are highest among younger (school-)age groups [18–22], so that if age-structured transmission is determined entirely by

contact frequency, we expect infection to be concentrated in these younger age groups. The relatively low levels of clinically presenting COVID-19 cases in children and teenagers therefore suggests that transmission is driven by more complex age-stratified heterogeneities beyond social contact patterns.

Although studies of social contacts are increasing in number [18, 20–27], they tend to be time consuming to conduct and therefore have been limited to a number of exemplar countries. Prem *et al.* (2017) extrapolated these contact matrices to 152 countries based on a range of social and demographic data [19]. This generates a publicly available set of country-specific contact matrices whose entries correspond to the expected total number of age-stratified contacts per day for individuals aggregated into 5-year age classes [19]. One study has already used these estimated contact patterns to model COVID-19 in China, although the potential for other age-specific heterogeneities was not addressed [28].

In this paper, we define a model of the early age-structured transmission dynamics in China which incorporates the country-specific social contact matrix (estimated by Prem *et al.* [19]) as well as components defining age-specific susceptibility profiles, probabilities of developing symptoms, and transmission rates from subclinical cases. Our model formulation is sufficient flexible to account for a range of different assumptions relating to the drivers of the observed age-structured heterogeneity; we estimate parameters to match age-structured data from the outbreak in China based on two different sets of assumptions which significantly simplify the model:

1. A null model where transmission is purely related to contact pattern. This would be the default assumption for any novel pathogen, but for COVID-19 relates to when all infected individuals are equally infectious independent of symptoms.

2. A model where there is negligible transmission from asymptomatic infections, such that younger age-groups contribute less to overall transmission.

We then combine our fitted transmission parameters with Prem *et al.*'s estimates of the contact matrices in the other 151 countries to generate an estimate of an age-structured transmission structure for each of those countries. The basic reproductive ratio can then be estimated from these transmission structures to give us an indication of the spreading potential of a COVID-19 outbreak in each of these countries.

## Materials and methods

We develop an age-structured transmission model based upon an age-structured contact matrix and allowing for age-stratified variation in susceptibility, probability of clinical symptoms, and reduction in transmission by asymptomatic (and therefore undetected) cases relative to clinically symptomatic (and hence detectable) ones. Full details of the model structure are given in S1 Appendix.

We parameterise the contact matrices for each country using the age-dependent contact rates estimated by Prem *et al.* [19]. The age-stratified susceptibility, symptomatic probability and asymptomatic transmission profiles need to be inferred from data. Given the large number of parameters involved (three times the number of age classes), we only carry out fits for a few specific assumptions, corresponding to different factors which can drive age-stratified heterogeneity. An important assumption throughout this work is that these age-stratified profiles are the same from population to population; we feel this is justified since they are likely to be driven by physiological mechanisms which are unlikely to be population-specific. In particular, we assume that detection is closely correlated with showing symptoms, so that the age-

structured distribution of reported cases reflects the age-structured distribution of symptomatic cases.

For reference, we first consider a simple null model in which all age groups are equally susceptible and all infected individuals (both symptomatic and asymptomatic) transmit at the same rate. That is, we assume that the next-generation matrix (which describes the number of new infections in each age-group generated by a case in a given age-group [5]) is directly proportional to the estimated contact matrix for China; such that contact patterns within the community are the only source of age-structured heterogeneity. The scaling of the contact matrix is performed such that the next-generation matrix generates a basic reproductive ratio of 2.4, in line with current literature. This scaling is applied to all country-specific contact matrices to generate their corresponding transmission matrices and associated basic reproductive ratio. (To match this model to the reported age-distribution of cases in China [4] the age-dependent probability of symptoms can be modified, which in turn determines the pattern of detection although leaves the epidemiology untouched).

We contrast the null model with one in which transmission from asymptomatic cases is negligible, and observed age-structured hetereogeneity is driven by a combination of age-dependent susceptibility and probability of symptoms. The latter has already been identified as of particularly importance, with secondary attack rates for children potentially similar to those among adults, despite the much smaller number of clinically presenting cases [29]. In this model age-structured heterogeneity in susceptibility profile or in the symptomatic probability have identical effects, and so we can fit a single set of parameters capturing both possible effects. Again, we fit this simplified model using epidemiological data from China [4] and the estimated China-level contact matrix [19], and scale the result such that the next-generation matrix generates a basic reproductive ratio of 2.4. The age-dependent parameters and scaling can then be combined with the estimated country-level contact matrices from the other countries in Prem *et al.*'s study [19] to produce age-stratified next-generation matrices for these countries, and hence an estimate of the country-specific basic reproductive ratio. We explain this calculation in detail in S1 Appendix.

All of the code used to implement our model and generate the results in the next section is available at https://github.com/JBHilton/hilton-keeling-estimating-R0.

## Results

In what follows we use an estimate of $R_0 = 2.4$ for the basic reproductive ratio of COVID-19 in China, consistent with values estimated in the literature so far ($R_0 = 2.2$ [1], $R_0 = 2.3 - 2.6$ [11], $R_0 = 2 - 2.7$, [10], $R_0 = 2.35$ [7], $R_0 = 3.11$ [9]), although we note that other estimates of $R_0$ in China can be accommodated by a linear rescaling of all the predicted values.

In Fig 1(a) we present the results of the null-model in which susceptibility is age-independent and all infectious individuals (both symptomatic and asymptomatic) have the same transmission rate(which can be matched to the age-structured case data from China by allowing the probability of displaying symptoms and therefore being detected to vary with age). We plot the estimated $R_0$ for each of the 152 countries included in Prem *et al.*'s study on a map of the world, with map colours grading from blue (low values of $R_0$) through yellow (intermediate values) to red (high values of $R_0$). Fig 1(b) contrasts this with the estimated basic reproductive ratios based on the age distribution of the first 44,672 confirmed cases in China, as reported by China's Novel Coronavirus Penumonia Emergency Response Epidemiology Team in China CDC Weekly [4]. Our numerical estimates of the basic reproductive ratio for each of the 152 countries are given in S1 Appendix.

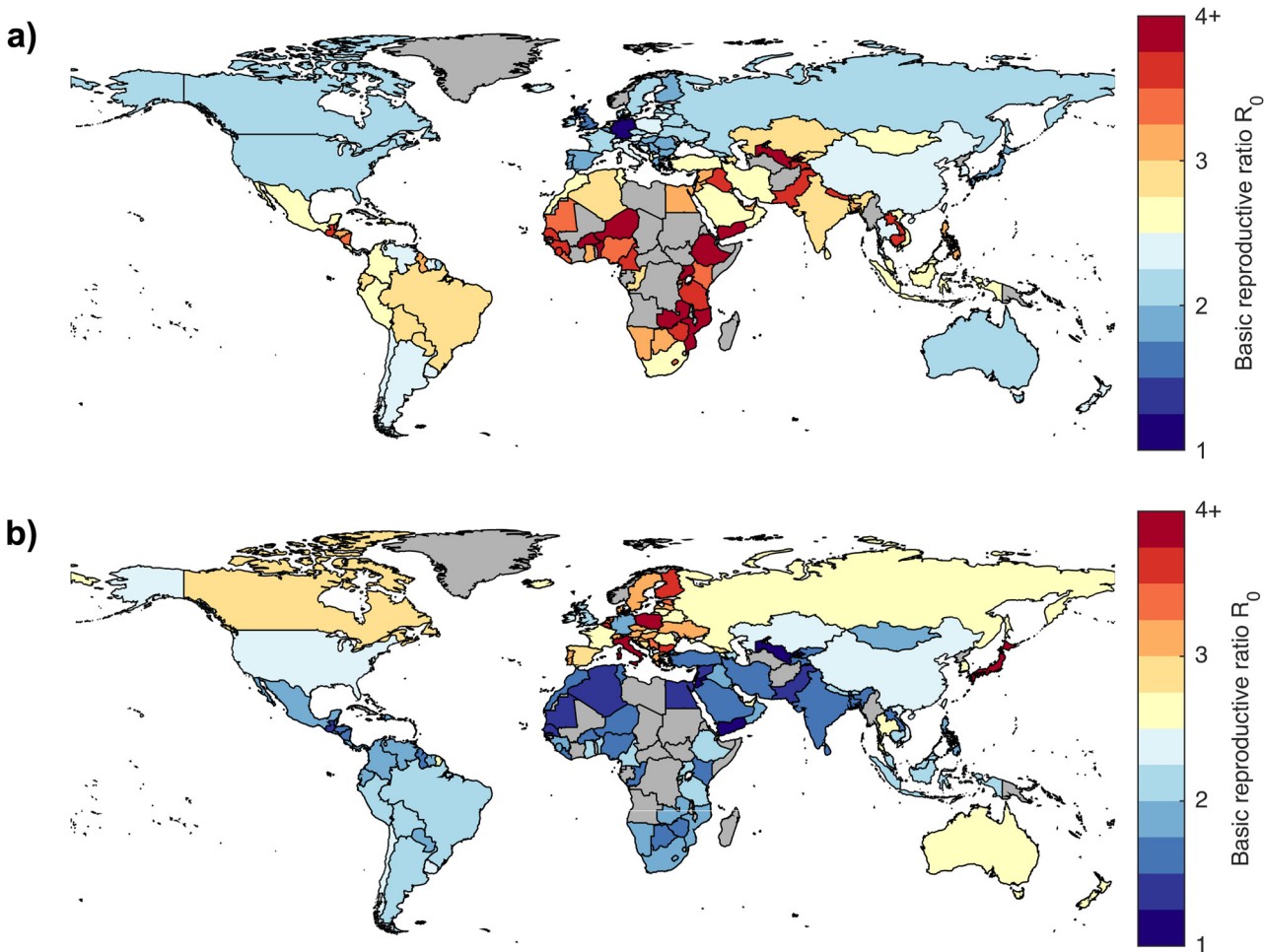

**Fig 1. Basic reproductive ratio by country.** (a) Estimated basic reproductive ratio for each country assuming contact structure only; (b) estimated basic reproductive ratio for each country based on the China CDC case data [4]. Gray countries are those not included in Prem *et al.*'s study [19].

To interpret these findings, in Fig 2(a) we plot the two sets of basic reproductive ratios shown in Fig 1, contrasting the null-model (x-axis) with the model in which there is negligible transmission from asymptomatic infections (y-axis). The assumptions underpinning the latter model lead to an increase in the amount of variation in basic reproductive ratio by country, and in particular can generate much larger basic reproductive ratios. For the null model (or when both symptomatic and asymptomatic infections are equally infectious), the variation in basic reproductive ratio is driven by the variation in average intensity of contacts by country. In contrast, when infection is driven by the age-dependent pattern of symptomatic cases, contact patterns involving members of highly symptomatic age classes becomes particularly important—generating a core-group within the population. For COVID-19, this means that we see higher basic reproductive ratios in countries with older populations, since this generally leads to contacts involving older individuals being more common. In countries with comparatively younger populations, contacts involving older individuals are far less common and so the capacity of the infection to spread is reduced relative to the null model.

This principle, that the underlying age-structure of the population drives the estimated basic reproductive ratio, is illustrated in Fig 2(b). Here we compare population pyramids of Niger (which is predicted to have the highest reproductive ratio under the null model) and

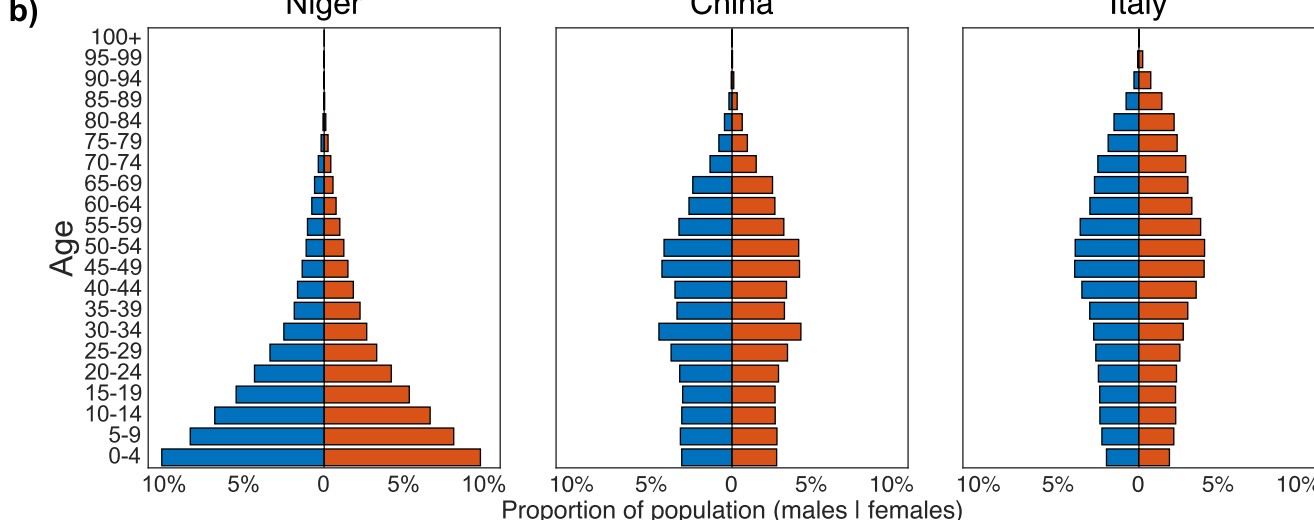

**Fig 2. Effect of using age-specific susceptibility/symptomatic probability and underlying population pyramids.** (a) Basic reproductive ratio estimates based on age-specific susceptibility or symptomatic probability estimated from China CDC Weekly data versus estimates without age-dependent susceptibility or symptomatic probability. (b) Population pyramids for Niger, China and Italy—China being our reference case (both $R_0$ values equal to 2.4), Niger having the highest $R_0$ in the null model, and Italy having the second highest $R_0$ in the age-specific susceptibility model. The highest $R_0$ attained in the age-specific susceptibility/symptomatic probability model is in Monaco, but since Monaco's small population is likely to make it an outlier we focus on Italy as an extreme case in the main cluster of ratios. Germany is also labelled in Figure (a); although it has a comparatively small $R_0$ under both sets of assumptions, the proportional change from 1.22 in the null model to 1.99 based on the China CDC data is almost as dramatic as that seen for Italy (2.44 to 4.18). Data from [30].

Italy (which is predicted to have one of the highest reproductive ratios under the second model) in comparison to China (which is the foundation of the parameter values). The population pyramid of Niger is dominated by young children; China has a relatively stable age-structure although there are more individuals in 30-54 age classes than in younger age-groups; the pyramid for Italy shows even fewer children and substantial proportions into older age-classes. We therefore postulate that it is the interaction between the population pyramid and the age-structured probability of symptoms (and hence significant transmission) that largely drives the scaling of the basic reproductive ratio.

This relationship between the population pyramid and the basic reproductive ratio is seen to hold for all countries investigated (Fig 1). In Fig 1(a), we observe that in the null model transmission is generally low in many European countries as well as in South Korea and Japan, and high in many African countries, consistent with the differences in daily number of contacts predicted by Prem *et al* driven by the proportion of children. However, Fig 1(b) shows that when age-specific susceptibility or symptomatic probability is taken into account, the pattern of infectious potential by country is generally reversed. We then expect to see higher transmission in Eastern Europe (including Italy which had the largest number of cases in Europe in mid March 2020) and Japan, and reduced transmission across Africa, central America, the Middle East and India.

## Discussion

Here we have developed a flexible model for age-dependent transmission of SARS-CoV-2 with four forms of heterogeneity: an age-structured contact matrix dependent on the behaviour of the host population; age-dependent susceptibility; age-dependent symptomatic probability; and transmission profiles dependent on physiological response to infection (symptomatic vs asymptomatic). By exploiting a previously-estimated synthetic contact matrix and age-stratified data, and by focusing on the specific case where undetected transmission is negligible, we were able to estimate these age-dependent profiles based on the first 44,672 cases in China [4]. We then combined these estimated profiles with estimates of age-stratified contacts in 151 other countries to generate transmission matrices for these countries from which we can also estimate the scale of basic reproductive ratios in each country relative to China.

We explored two competing models. The null model is purely based on the frequency of age-stratified contacts, and would be the default assumption for any novel pandemic. Given the flexible way in which our model is constructed, allowing an age-dependent probability of displaying symptoms yet symptom independent transmission (asymptomatic and symptomatic infections transmit equally) leads to the same basic reproductive ratio. The second model assumes that detected cases represent a random sample of symptomatic infections and therefore provide an unbiased measure. It is also assumed that the only symptomatic infections are responsible for the overwhelming majority of onward transmission. This allows us to reliably match to the early outbreak data in China [4]. Moreover, the second model allows us to account for age-structure either through age-dependent susceptibility or age-dependent probability of symptoms, or any mixture of the two. We demonstrated that taking such age-specific factors into account results in substantially different predictions of transmission intensity by country relative to a purely contact-based null model; countries with older populations are at substantially higher risk than countries with younger populations.

There are two main limitations to our predictions. The first is the accuracy of the estimated contact matrices; although there are known issues (as discussed in [19]) they remain our best estimate of age-structured contacts to date across many regions of the world. Unfortunately, not all countries have an associated mixing matrix, as many countries (predominantly in

Africa) do not have the underlying demographic data recorded that supports the generation of this matrix. This is clearly problematic since mortality from infectious disease is disproportionately concentrated in these countries [31]. Secondly, to estimate the basic reproductive ratios mapped in Fig 1(b) we chose to focus specifically on the case where transmission from undetected cases was negligible. This assumption makes it possible to analytically determine age-structured parameters, and reduces the need to separate the effects of susceptibility and probability of displaying symptoms. If asymptomatic infections transmit as strongly as symptomatic cases, then the country-specific reproductive ratio is expected to be closer to Fig 1(a).

Finally, it is worth stressing that these projections only capture the early phase of the outbreak in the absence of controls. Non-pharmaceutical interventions (contact tracing, self isolation and movement controls) can substantially reduce the infection's reproductive ratio, with their effectiveness heavily dependent on the pre-intervention basic reproductive ratio, the proportion of contacts traced, and the timing of isolation and movement controls [8, 29, 32–34].

## Supporting information

**S1 Appendix. Model description, calculations, tables of $R_0$ by country.**
(PDF)

## Author Contributions

**Conceptualization:** Matt J. Keeling.

**Data curation:** Joe Hilton, Matt J. Keeling.

**Formal analysis:** Joe Hilton, Matt J. Keeling.

**Investigation:** Joe Hilton, Matt J. Keeling.

**Methodology:** Joe Hilton, Matt J. Keeling.

**Visualization:** Joe Hilton, Matt J. Keeling.

**Writing – original draft:** Joe Hilton, Matt J. Keeling.

**Writing – review & editing:** Joe Hilton, Matt J. Keeling.

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
