## [Decision Letter · Decision Letter 0]

27 May 2020

Dear Mr Hilton,

Thank you very much for submitting your manuscript "Estimation of country-level basic reproductive ratios for novel Coronavirus (SARS-CoV-2/COVID-19) using synthetic contact matrices" for consideration at PLOS Computational Biology. As with all papers reviewed by the journal, your manuscript was reviewed by members of the editorial board and by several independent reviewers. The reviewers appreciated the attention to an important topic. Based on the reviews, we are likely to accept this manuscript for publication, providing that you modify the manuscript according to the review recommendations.

It would be great to update the github repository to include R0 estimates, as suggested by Reviewer 2.

Sincerely,

Jennifer A. Flegg

Associate Editor

PLOS Computational Biology

Virginia Pitzer

Deputy Editor

PLOS Computational Biology

[LINK]

It would be great to update the github repository to include R0 estimates, as suggested by Reviewer 2.

Reviewer's Responses to Questions

**Comments to the Authors:**

Reviewer #1: Thank you for the opportunity to review “Estimation of country-level basic reproductive ratios for novel Coronavirus using synthetic contact matrices.” The authors estimate the basic reproductive ratio of SARS-CoV-2 for 151 countries and illustrate the need to better understand age-specific transmission patterns of SARS-CoV-2 (specifically age-dependent susceptibility, age-dependent symptomatic probability) and the transmission profiles dependent on physiological response to infection. The introduction, methods, and results are well-written, clear, concise, and appropriate to evaluate the objective. This paper adds important nuance to the growing scientific literature and estimation of the basic reproductive ratio. It was a nice read.

I do not have any revisions except that I noticed a handful of typos that I suspect if accepted will be caught prior to publication. For example, lines 154, 189 - probability instead of probably; line 153 - add "the".

Reviewer #2: The review is attached.

**Have all data underlying the figures and results presented in the manuscript been provided?**

Reviewer #1: Yes

Reviewer #2: No: The github repository linked to contains the scaling factors under both models as CSV files, but not the $R_0$ values used to create the maps. It would be worth the authors including the estimated $R_0$ values so that users do not have to apply the scaling themselves.

PLOS authors have the option to publish the peer review history of their article (what does this mean?). If published, this will include your full peer review and any attached files.

Reviewer #1: No

Reviewer #2: Yes: Samuel Clifford
---

## [Editor Report · Decision Letter 1]

8 Jun 2020

Dear Mr Hilton,

We are pleased to inform you that your manuscript 'Estimation of country-level basic reproductive ratios for novel Coronavirus (SARS-CoV-2/COVID-19) using synthetic contact matrices' has been provisionally accepted for publication in PLOS Computational Biology.

Best regards,

Jennifer A. Flegg

Associate Editor

PLOS Computational Biology

Virginia Pitzer

Deputy Editor

PLOS Computational Biology

The comments from both reviewers have been addressed fully.

---

## [Editor Report · Acceptance letter]

26 Jun 2020

PCOMPBIOL-D-20-00579R1 

Estimation of country-level basic reproductive ratios for novel Coronavirus (SARS-CoV-2/COVID-19) using synthetic contact matrices

Dear Dr Hilton,

I am pleased to inform you that your manuscript has been formally accepted for publication in PLOS Computational Biology. Your manuscript is now with our production department and you will be notified of the publication date in due course.

With kind regards,

Matt Lyles
